# Information sharing challenges in end-of-life care: a qualitative study of patient, family and professional perspectives on the potential of an Electronic Palliative Care Co-ordination System

Holly Standing [1], Rebecca Patterson,[2] Mark Lee,[3] Sonia Michelle Dalkin [4]
Monique Lhussier,[4] Angela Bate,[1] Catherine Exley,[5] Katie Brittain[5]

For numbered affiliations see end of article.

**Correspondence to**
Dr Holly Standing;
holly.standing@northumbria.ac.uk

## ABSTRACT

**Objectives** To explore current challenges in interdisciplinary management of end-of-life care in the community and the potential of an Electronic Palliative Care Co-ordination System (EPaCCS) to facilitate the delivery of care that meets patient preferences.

**Design** Qualitative study using interviews and focus groups.

**Setting** Health and Social Care Services in the North of England.

**Participants** 71 participants, 62 health and social care professionals, 9 patients and family members.

**Results** Four key themes were identified: information sharing challenges; information sharing systems; perceived benefits of an EPaCCS and barriers to use and requirements for an EPaCCS. Challenges in sharing information were a source of frustration for health and social care professionals as well as patients, and were suggested to result in inappropriate hospital admissions. Current systems were perceived by participants to not work well—paper advance care planning (ACP) documentation was often unavailable or inaccessible, meaning it could not be used to inform decision-making at the point of care. Participants acknowledged the benefits of an EPaCCS to facilitate information sharing; however, they also raised concerns about confidentiality, and availability of the increased time and resources required to access and maintain such a system.

**Conclusions** EPaCCS offer a potential solution to information sharing challenges in end-of-life care. However, our findings suggest that there are issues in the initiation and documentation of end-of-life discussions that must be addressed through investment in training in order to ensure that there is sufficient information regarding ACP to populate the system. There is a need for further qualitative research evaluating use of an EPaCCS, which explores benefits and challenges, uptake and reasons for disparities in use to better understand the potential utility and implications of such systems.

### Strengths and limitations of this study

► Large participant group from various stakeholder groups including health and social care practitioners, patients and carers.
► This is the first study to include the experience of social workers and of a coroner.
► Semistructured interviews allowed us to collect rich descriptions of participants' experiences and perceptions of information sharing in end-of-life care and of an Electronic Palliative Care Co-ordination System (EPaCCS).
► The study was conducted by a team of experienced researchers from a range of disciplinary backgrounds.
► Data are from the preimplementation phase of an EPaCCS Trial, so findings are not drawn from direct experience of the system.

## INTRODUCTION

The development of end-of-life care guidance in recent years supports attempts to allow dying persons more control over their end-of-life experience, through engagement of patients in advance care planning (ACP).[1] ACP has been defined as 'a process that supports adults at any age or stage of health in understanding and sharing their personal values, life goals and preferences regarding future medical care'. The goal of ACP is to help ensure that people receive medical care that is consistent with their values, goals and preferences during serious and chronic illness.[2] ACP can include anyone involved in the provision of the patient's care, including nurses, doctors, care home managers and family members. Documents completed as part of ACP can include: a do not attempt

cardiopulmonary resuscitation order, emergency health-care plan (EHCP) and a written summary of the patients care preferences.[3] Paper versions of this documentation are completed and held by the patient to present to the health and social care professionals they encounter.[4] ACP can facilitate the delivery of care, which better reflects the patient's personal preferences,[1] and can have positive impacts on the person at the end-of-life and their family.[5] However, evidence suggests that the completion and use of ACP documentation is inconsistent.[6 7]

The delivery of high-quality end-of-life care is an inter-professional endeavour, where the resources of various health and social care professionals must be combined for the benefit of the patient.[8] To achieve this, efficient inter-professional communication and information sharing across health and social care professionals involved in community end-of-life care is essential,[9] but presents many challenges.[10–12] Accessibility of patient information can be limited, particularly for services who may only be episodically involved in end-of-life care but that offer crucial and timely support, such as paramedics, out-of-hours doctors, admiral nurses (dementia specialist nurses), care home staff and social workers.[13 14] Poor (or lack of) access to patient information can effect professionals' confidence in managing end-of-life patients,[13 14] family members satisfaction with care[15] and patient outcomes.[5] Sharing of patient information is made more challenging by inconsistent documentation practices and systems that vary across organisations and professional boundaries.[16]

Electronic record systems offer a potential solution to these information sharing challenges.[17] A Framework for Action published by the National Information Board in 2014 details 2020 as a deadline for digital, real-time and interoperable health records to be rolled out within the National Health Service (NHS).[18] Electronic Palliative care Co-ordinating Systems (EPaCCSs) are example of such an innovation. EPaCCS is an umbrella term covering a number of different electronic solutions that aim to capture patient wishes and preferred place of death and improve co-ordination of care in real time, through enabling the sharing of information across health and social care services. A 2013 Department of Health pilot evaluation in England estimated that EPaCCS could save £133 200 in hospital admissions per annum per 200 000 patient population.[19] There are a number of versions of EPaCCS currently in use in the UK including the Key Information Summary (KIS) in Scotland,[20] Coordinate My Care in London,[21] South West EPaCCS[22] and Leeds EPaCCS.[23] Like many areas of the UK, the North East England does not yet have an EPaCCS in place. Research from other regions indicates that professional uptake and patient registration on such electronic systems is currently variable.[20 22] A systematic review found that 55%–79% of patients with an EPaCCS died in their preferred place, higher than the general population.[24] The Scottish KIS has been suggested to be particularly beneficial for General Practitioners (GPs) working out-of-hours.[25] Challenges surrounding the use of an EPaCCS include professional reluctance to initiate end-of-life discussions,[6 26] increased burden from data entry for the primary users,[22] data ownership, funding and consent.[27]

The aim of our study was to explore attitudes towards the potential of an EPaCCS solution for improving information sharing and coordination in end-of-life care within the North East of England. We provide an in-depth study of the professional and organisational factors that promote or inhibit the acceptability, usefulness and integration of collaborative care planning across health and social care into service delivery and everyday practice.[28–30]

## METHODS

This study employed qualitative methods, involving semi-structured interviews and focus groups with patients, bereaved family members, and health and social care professionals.

### Participants

A heterogeneous purposive sampling approach was employed to ensure representation from all health and social care professional groups who have a potential role in end-of-life care, and that a variety of perspectives and experiences of end-of-life care were sampled.

### Data collection

An interview schedule was developed by HS, RP and KB that included: current approaches to end-of-life care management, access to patient data and attitudes towards introduction of an EPaCCS (including perceived benefits and challenges), an example interview schedule is provided in the online supplemental material. A video demonstration of a draft version of the proposed EPaCCS was also used as a prompt during clinician interviews. The interview schedule evolved during the data collection process to allow iterative data analysis and further exploration of themes derived from the data. Informed consent was given by all participants prior to taking part. Interviews were digitally recorded.

Participants were recruited from an area within the North East of England where an EPaCCS was being trialled, data presented are from the preimplementation phase of the trial. Patient and family participants were recruited through GP practices, practice staff acted as gatekeepers and sent information packs to patients on their palliative care registers and next of kin of deceased patients, those interested in participating returned a consent-to-contact form in a prepaid envelope. Clinicians were recruited through local clinical networks, calls for volunteers circulated via email, attendance of the study team at local commissioning meetings. Data collection took place from March to October 2018, interviews and focus groups were conducted at the participants' place of work or home, at university or community venues or over the telephone, according to participant preference. Interviews and focus groups were conducted by HS, RP and KB. Clinician data were generated through 24 interviews

Table 1 Participant demographics

| Health and social care professional role | Number |
| --- | --- |
| General Practitioner (GP) | 7 |
| Out-of-hours GP | 2 |
| Nurse (including: Macmillan (cancer care nurse), Marie Curie (terminal care nurse), care home, district, rapid response, admiral nurse (dementia care nurse)) | 21 |
| Formal carer (care home) | 3 |
| Paramedic | 10 |
| Social worker | 6 |
| Pharmacist | 4 |
| Hospital doctor | 4 |
| Other supporting professions (including: care coordinator, Macmillan manager, physiotherapist, coroner, palliative care team) | 5 |
| **Patient/family** | |
| Patient | 7 (4 female, 5 lived alone, 6 60+ years old) |
| Family | 2 (both females, 60+) |
| **Total** | **71** |

and 5 focus groups. Two of the five focus groups were conducted with a mix of health and social care professionals including hospital doctors, pharmacists and out-of-hours GPs (n=9 and n=10). The remaining three focus groups were conducted with single-profession groups: care home staff (n=5), social workers (n=5) and nurses (n=9). Cancer was the primary diagnosis for all patient interviews and family interviews. Table 1 details participant demographics. Clinician interviews ranged from 28 to 93 min, focus groups from 39 to 63 min and patients and family member interviews from 22 and 80 min. Field notes were collected before, during and after data collection. Data have been pseudonymised to protect participant identity.

### Data analysis

Audio recordings were transcribed verbatim by an independent transcription company; transcripts were checked for accuracy by (HS, RP and KB). The data management software NVivo V.12 was used to develop and refine a coding scheme.

We adopted an iterative approach to data analysis. Data collection and analysis ran concurrently throughout the study and analysis of early transcripts informed the refinement of the interview schedule for later interviews. The five-step process of thematic analysis[31] was adopted to develop and refine codes and themes from the data. Field notes taken during data collection were used in

data analysis to enhance the reflexive process. Reflexivity is the process of accounting for the situatedness of the researcher within the research and the potential effect of this on the data collected and interpretation.[32] Credibility measures included a process of continuous review and discussion of themes at group analysis sessions, and at wider team meetings. Three researchers (HS, RP and KB) independently coded 20% of the transcripts before coming together to compare codes and discuss discrepancies.

### Patient and public involvement

The study was supported by a patient and public involvement representative who contributed to the design and development of the study, advised on recruitment processes, and participant information and consent materials.

### FINDINGS

Four themes were identified through data analysis: information sharing challenges; information sharing systems; perceived benefits of an EPaCCS and barriers to use and requirements for an EPaCCS. These are presented in turn below, with illustrative quotations.

### Information sharing challenges

Participants were keen for end-of-life patients in the community to be managed at home and avoid inappropriate hospital admissions where possible; however, poor communication and information sharing between services was a barrier to achieving this goal. Paramedics and out-of-hours GPs may be alerted by call handlers that the patient they are attending is at the end-of-life, but often were going in 'blind' as they had no prior access to the patient's condition or care preferences. Without access to up-to-date patient information, paramedics felt that in such situations they had no choice but to admit the patient:

> The number of times I've said, 'They are going to have to go in. I don't want to take them in, but I don't have a choice here, because I don't have the information.' If you don't have the information, you can't make an informed decision, so to leave them at home would be the wrong thing to do, even though you know morally you'd like to. (Paramedic6).

Inappropriate hospital admission of end-of-life patients was also a source of frustration for GPs who suggested that these could be avoided if they were contacted for advice:

> I suppose the admissions where they're palliative are the ones that I get annoyed about [...] they get admitted, and if somebody had asked you, you know, 'What should we do with this lady?' [we could advise on management but] that's not always easy because we don't work every day. We are obviously not working in the evenings or the weekends[...] (GP6).

Indeed, the continuity of care afforded by the GP role means they often have established relationships with end-of-life patients and would most likely have access to detailed information regarding their condition, history and care preferences, which should inform decision-making in any circumstance. However, as in-hours GPs are not available 24/7, there is a need for patient information to be stored in a manner that is easily accessible to all services who may require it. Paramedics or out-of-hours GPs are unlikely to have had prior contact when called out to an end-of-life patient, and would benefit from access to greater patient information to facilitate their management of such patients, and potentially avoid hospital admissions.

Poor communication and information sharing could have negative impacts on the patient and their family. Where patient information is unavailable, and services were going in 'blind' the burden of information sharing falls on the patient or their family.

> Whatever doctor I would see, they wouldn't know a lot about me, so I'd have to constantly start from the beginning […family] couldn't come into the meetings with me because they'd get so upset because they'd watch me explain everything from the start. I think that was worse than hearing any bad news (Patient5).

Having to repeatedly reiterate their story could be distressing for the patient and their family, and could also damage the professional patient relationship; as the patient above describes this constant repeating of her story led her family to disengage from encounters with health and social care professionals.

Lack of information sharing mean it is not always clear who should be responsible for verifying death and issuing the death certificate. This could result in the death being designated as 'unexpected' and the coroners being inappropriately involved, causing unnecessary delays and distress to the family:

> when they've gone home expecting that [the patient will die] and then all of a sudden somebody says, 'We don't know who can issue the certificate because the GP hasn't seen them because it was the out-of-hours doctor [who came]…' and it takes hours to sort it out. (Coroner1)

### Information sharing systems

Information about patients at the end-of-life is primarily stored in two formats: (1) paper-based ACP documents which are held by the patient and (2) isolated patient record systems (as it currently stands) versus an integrated system that can be updated and available to all professionals (and patients in 'real time').

Paper ACP documents facilitate information sharing; however, their usefulness is largely dependent on the patient, or their family, having the documents ready to hand or easily accessible and up to date. Many participants indicated this was often not the case:

> you're meant to have a bright yellow envelope on your mantle or on your sideboard. So, if you go in as a doctor, or if you go in as an ambulance crew, you know it's there, and it's bright, illuminous yellow, but the number of folks, 'Oh, I've got one of those,' and it's under four piles of stuff, or it's in their bedside cabinet. It's like, 'Well, no-one's going to respect that,' because no-one's going to spend 10 minutes looking for it if you've arrested. They're going to start resus. (Out-of-hours GP1).

If the documentation is not available, either misplaced or forgotten, then health and social care professionals have to proceed as if it did not exist.

Having a standardised location for the documentation in the patient's home (such as the mantelpiece, for example) may improve access to this information. However, keeping the documentation in a visible location may also be a potential source of distress for families. While some patient participants were comfortable having theirs visible, others chose to keep them hidden as they acted as a constant reminder of the imminent death.

> I had put the yellow envelope in my knicker drawer […] They said, 'Put it up there.' [mantel piece] So, I said, 'No, I'm not putting it up there. I don't want to look at that all the time.' (Bereaved family2).

In addition to paper versions of ACP documentation where available, health and social care services use electronic patient record systems to record and manage the information that they collect. These systems tend to work in isolation, and are generally not interoperable, meaning there is no simple way to share information with others. Many patients assumed that their information was being shared, and they did not necessarily perceive services as discrete organisations, rather part of a united healthcare system. Although patients recognised that protecting patient confidentially was important, the benefits of sharing information were perceived to significantly outweigh these concerns.

> What if you travel? What if you're visiting round the country? You don't know what's going to happen […] I went to London a couple of times when I was ill. What happens if something happens to me down there? How long would it take them to find anything out about me? In the meantime, what's happening to me? (Patient5).

### Perceived benefits of an EPaCCS

Participants identified several potential benefits of adopting an EPaCCS. NHS electronic patient records systems were perceived by the majority of participants, both patient and professional, as being extremely disjointed and often outdated.

> I certainly think there should be a national set-up. Everybody should be reading from the same sheet, yes. (Patient6).

Patients welcomed a system that would facilitate better transfer of information between services so that they would no longer have to act as the conduit for passing information between services by repeating their story.

Professionals indicated that an EPaCCS may save time and facilitate the provision of care that was better aligned with patients' wishes. An EPaCCS could potentially allow multiple users to simultaneously access and update a patient's record, ensuring changes to the patient's condition and/or care plan are immediately accessible to all professionals involved in their care.

> I think the thing is, when it's a paper copy there's only one actual current live copy that is up to date […] the electronic copy means multiple users could be using it at the same time. (Nurse6).

An EPaCCS was suggested to be particularly beneficial to services without sustained relationships with end-of-life patients who were often 'information poor'. Improved access to patient information could potentially increase the confidence of these services in delivering end-of-life care.

> You do get very limited information and you're going into a person to talk to them about what care they need with very limited information. And sometimes when they are end-of-life, the last thing they want to talk about is how they're going to die and why they're going to die (Social Worker5).

### Barriers to use, and requirements for an EPaCCS

Professional participants did not express universal enthusiasm for the introduction of an EPaCCS. Paramedics and out-of-hours GPs were generally positive about the potential of the system, as discussed above; however, GPs were less enthusiastic; the system was not necessarily perceived to offer improvements over current approaches to information storing and recording. GPs expressed concern about implications of the system for their workload; GPs are often the most 'information rich' meaning responsibility for inputting data into the system would fall on them. If the proposed EPaCCS was not fully interoperable with existing systems it would result in duplication of tasks and increased demands on GP time.

> If it's [a] separate [system] then you'd be duplicating because you'd need to add it to the system wouldn't you? […] You'd find GPs wouldn't engage with it, because it would just be an extra thing they have to do. (GP3).

The implications of a shared EPaCCS on maintaining data protection and the security of patient information wwere a major concern for many of the professional participants.

> We're still recovering from a cyber-attack a year ago, and there are still some documents that we can't

open […] so it has to be encrypted to a level that it's safe (Hospital doctor3).

Such concerns are understandable given recent security threats[33] and scandals involving misuse of personal data.[34] However, this did not appear to be a concern for patients, many of whom assumed that data sharing practices already occurred and viewed moves to facilitate information sharing through electronic records with the aim of improving care as welcome and necessary progress.

Incorporating the informational requirements of all health and social care groups into one system was highlighted as a challenge; different roles have different informational needs. For example, nurses indicated that lists of patient medication are essential information for their practice to ensure they are not prescribing contraindicated medications. Catering for individualised requirements such as these risks the EPaCCS becoming overwhelming and challenging to navigate.

> It might just slow people down. It will take longer actually to get through cases because you have to read through them [patient notes] (Out-of-hours GP1).

Having to sift through irrelevant information to find what is germane to their role may become burdensome. An EPaCCS with extensive information about the patient and limited in its ability to filter efficiently may not be preferable to paper ACP documentation being available on arrival.

Professionals who spend most of their time working remotely, doing home visits, may gain the most benefit from an EPaCCS.

> electronic information, that would solve the main problems that we have, which is lack of information at times. So yes, that would be great. (Paramedic5).

However, an EPaCCS needs to be accessible to these professionals. If an EPaCCS cannot be accessed and updated by those working in the community, the information contained within will quickly cease to be up to date. Successful uptake of an EPaCCS is also dependent on the technology available to access the system. Electronic devices available to those working in the community were reported to often be of low specification, raising doubts about likelihood that they would be able to support accessibility.

> It also has to be an 'on the go' portal, you were saying how busy you are and so it has to work on a tablet or on an iPad because a lot of this work is being inputted in the community. You can't rely on someone going back to their office, after they've done five house calls or five social work visits or five whatever […] It has to be a real-time and easy to access. (Hospital Doctor2).

A need for ongoing support for the EPaCCS following the implementation period was highlighted. Participants expressed concern that investing time learning a new

system would be wasted if ongoing infrastructure was not available to support its continued use.

> I've done many projects before within health, and there's a massive emphasis placed at the beginning of every project and they put it on. Then, afterwards, staff move away, because it's not part of their project and we're suddenly left with it. The system is changing and evolving, but there's not always the infrastructure in place to support us to update it. I think that's a real concern. (Nurse19).

Implementation of an EPaCCS was not necessarily perceived to solve the challenges in interdisciplinary management of end-of-life care alone. Access to ACP documentation will only be useful if the documentation itself is completed to a satisfactory standard, if insufficient detail is provided it cannot be used to inform decision-making. Paramedics and out-of-hours doctors indicated that ACP documentation that they encountered was of a variable standard, and these documents should be completed with these services in mind as the intended reader.

> I'm making a decision within 30 seconds I've walked through the door, 'Am I going to attempt to save a life or am I going to let them expire?' The only thing between me and setting off a treatment path is the end-of-life care plan, so it needs to make perfect sense. (Paramedic4).

## DISCUSSION
### Principle statement of findings
This study provides insight into current challenges in information sharing in end-of-life care, which were recognised as a barrier to effective end-of-life care, and the potential of an EPaCCS system to overcome these challenges. Inconsistent information sharing within and between health and social care services was a source of frustration for patients, paper based ACP documentation does not appear to be an effective system for recording and sharing patients' care preferences; often these documents are unavailable when required and are variable in quality. Patients welcomed the idea of an EPaCCS that facilitated sharing of their information between health and social care services, they did not report the same concerns about data protection and security issues that concerned clinicians. Both groups of participants perceived the potential for improvements in delivery aligned with patient's wishes from adoption of an EPaCCS. Perceived barriers to the successful implementation of an EPaCCS were focused on increased demand on time and the lack of infrastructure in place to support the system. Care must be taken to ensure that information contained within the EPaCCS does not become overwhelming. This is particularly important for emergency services, such as paramedics, who work under extreme time pressures limiting their ability to access and digest vast amounts of information. Introduction of an EPaCCS was not seen to resolve current challenges in management of patients at the end-of-life without investment into improving the frequency with which ACP documentation is completed and the quality of the information recorded. Paramedics in particular suggested their informational needs must be a priority to ensure ACP documentation is completed to a standard where it can be used to inform decision-making.

### Strengths and weaknesses
A key strength of this study is the large sample that includes a wide range of health and social care professionals involved in end-of-life care, the majority of previous qualitative studies have employed small samples,[35 36] and ours is the first study to our knowledge to include the perspectives of supporting professions such as social workers and coroners. In addition, our study included a greater number of participants from out-of-hours services than previous qualitative studies.[22 36] As the data were collected during the developmental preimplementation phase of an EPaCCS Trial, participants were only able to reflect hypothetically on the impacts of such a system on their practice rather than drawing on direct experiences of working with the system. In line with previous EPaCCS studies, a second limitation of the study stems from the recruitment process for patients and family members, which was, operationalised through GP practices. Staff thus acted as gatekeepers and may have been selective in who they referred for participation.

### Comparison with other work
There are only a limited number of qualitative studies exploring attitudes towards and use of EPaCCS.[22 35–37] Our findings support those of a recent systematic review that identified the burden of inputting data and Information Technology (IT) systems as the main challenge to implementation of EPaCCS.[24] GP participants expressed a reluctance to dedicate time to inputting data into the EPaCCS, this has been observed in previous studies where GPs have reported delegating much of this work to administrative staff.[22] However, there is high use of the Scottish KIS system despite reliance on GPs to populate.[20]

A key finding of our study was that introduction of an EPaCCS alone does not provide a solution to some of the current difficulties regarding interdisciplinary management of end-of-life patients in the community. An EPaCCS will only facilitate access to patient information, the utility of which is largely dependent on its quality; improving access to ACP documentation is not going to improve care if the information recorded within them is of poor quality or insufficient detail, or has not been completed. This demonstrates a need for investment in training for health and social care professionals in ACP discussions and documentation to instil confidence and improve quality. Inconsistencies in ACP documentation have also been observed within an Australian healthcare context.[38]

This article highlights concerns regarding the infrastructure required to support an EPaCCS including: demand on already stretched professionals, technology provision required to enable access and continued support to ensure the system is updated. Ensuring appropriate infrastructure is in place has emerged as a challenge in the implementation of coordinated records for end-of-life patients in both the UK and US contexts.[24 39] As this article focuses on the preimplementation stage of an EPaCCS Trial future research should explore the long-term uptake and impact of an EPaCCS to better understand the infrastructure required to stain such a system. Although there have been recent mixed methods studies conducted post implementation of an EPaCCS,[20 25 40] there is a need for further qualitative research exploring the benefits and challenges experienced while using the system for all stakeholders including in-hours and out-of-hours services, and in particular on the impacts of the experiences of patients and families. High and low users of the system should be explored to identify reasons for disparities and potential ways of promoting use of the system.

## Conclusions

EPaCCS may offer a potential solution to information sharing difficulties in the interdisciplinary management of end-of-life care. However, implementation of an EPaCCS alone will not improve management of patients at the end-of-life it will only facilitate the access to patient information. Introduction of EPaCCS must be accompanied by investment in training around communication skills focusing on the initiation of end-of-life discussions and effective completion of ACP documentation.

**Author affiliations**
[1]Department of Nursing, Midwifery and Health, Northumbria University, Newcastle upon Tyne, UK
[2]Research and Innovation Services, Northumbria University, Newcastle upon Tyne, UK
[3]St Benedict's Hopsice and Specialist Palliative Care Centre, Sunderland, UK
[4]Department of Social Work, Education and Community Wellbeing, Northumbria University, Newcastle upon Tyne, UK
[5]Population Health Sciences Institute, Newcastle University, Newcastle upon Tyne, UK

**Acknowledgements** We acknowledge the funder, the wider project team (led by Dr Kathryn Hall, clinical lead), Michele Spencer from North Tyneside Community and Health Care Forum for her input as patient and public representative and most importantly the participants who gave their time to the research.

**Contributors** All authors contributed to the design of the study. HS, KB and RP conducted the data collection and analysis. HS and KB drafted the manuscript. All authors revised the article critically for important intellectual content. All authors approved the submitted draft.

**Funding** This paper summaries independent research funded by Connected Health Cities

**Disclaimer** The views expressed are those of the authors and not necessarily those of Connected Health Cities.

**Competing interests** None declared.

**Patient and public involvement** Patients and/or the public were involved in the design, or conduct, or reporting, or dissemination plans of this research. Refer to the Methods section for further details.

**Patient consent for publication** Not required.

**Ethics approval** Ethical approval for this study was obtained from the Health Research Authority (REC reference: 17/LO/2100).

**Provenance and peer review** Not commissioned; externally peer reviewed.

**Data availability statement** Data are available upon reasonable request. Given the sensitive nature of this qualitative research project the raw data generated and analysed during the current study are not publicly available to preserve the anonymity of participants. However, we are willing to make portions of transcripts (with any identifying information redacted) available upon reasonable request. Interested persons should contact the corresponding author.

**ORCID iDs**
Holly Standing http://orcid.org/0000-0002-7806-8596
Sonia Michelle Dalkin http://orcid.org/0000-0002-3266-5926

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
