## [Reviewer comments · BMJ Open]

ARTICLE DETAILS

TITLE (PROVISIONAL)	Information sharing challenges in end-of-life care: a qualitative study of patient, family and professional perspectives on the potential of an Electronic Palliative Care Coordination System
AUTHORS	Standing, Holly; Patterson, Rebecca; Lee, Mark; Dalkin, Sonia Michelle; Lhussier, Monique; Bate, Angela; Exley, Catherine; Brittain, Katie

VERSION 1 – REVIEW

REVIEWER	Deb Rawlings Flinders University, Adelaide, Australia
REVIEW RETURNED	02-Mar-2020

GENERAL COMMENTS	Information sharing challenges in end-of-life care and the potential for an EPACCS: patient, family and professional perspectives Thank you for this interesting paper. I did however, find it difficult to reconcile two of the main points that you are making – that ACP processes do not work well (true), and that an electronic system would help with information sharing (true, if the information was available, and if all health professionals involved had access to the same program on their system and had time to use it). Abstract conclusion: You state that current systems for ACP do not work well, but it is hard to understand how an electronic version would be a solution to documentation that is fragmented and often of low quality (if completed at all). The solution surely needs to start with addressing ACP processes before putting them into a system that has yet to prove itself and that requires training and resources? Page 5. Line 13 Please explain what Admiral nurses are for those outside the UK Page 5. Line 50. Surely professional reluctance to initiate EOL discussions relates more to a lack of communication and lack of subsequent documentation generally, rather than a reflection of the use /uptake of EPaCCS? Page 6. Methods section. I found this to be lacking the details that are described in the proforma at the end of the paper. In particular recruitment approaches / population pool /sampling and reporting on participant demographics. Page 6. Line 52. Please detail numbers approached and numbers
---

	finally interviewed Page 7. Line 23. Please explain Macmillan and Marie Curie roles Page 7. Please detail which participants were interviewed and which were part of a focus group. Were the latter grouped via discipline and patient / family for example? Page 7. Line 58. Is there a reason that interview transcripts did not go back to participants for verification? Page 8. Line 12. Could you please expand on reflexivity and why it was important? Page 8. Line 18. Did you perform any cross-verification analysis? What was your level of agreement? Page 12. Line 6. Is the reader to assume that every patient at end of life in this region has an ACP? Page 17. Discussion. It would be interesting to see a summary (table?) of the findings by grouping eg, patients, paramedics, GPs. There are important considerations for each as well as competing interests. It would also be interesting to see a discussion on whether the implementation of an electronic patient case note system would in fact have any effect on ACP completion rates, quality of ACP documentation and so on. It is idealistic to think that a system like this would solve all communication between disciplines as it always is dependent on users imputing the information. Page 18. Line19. This is the first mention of recruitment concerns – not mentioned previously and an important consideration in study design Page 18. Line 52. Comparison with other work. You make important points here about limitations of EPaCCS in relation to EOL management. This section addresses most of my questions and is almost lost in the paper. Page 19. Line 42 Conclusion. This is pretty limited and should reflect more the findings from the study. Page 18 lines 52 onwards is (in part) more of a conclusion that represents your work (as above). I think that all points are here, the paper would benefit from some additional information (tables) and a rework of some sections.
--	---

REVIEWER	Anne Finucane University of Edinburgh UK
REVIEW RETURNED	05-Mar-2020

GENERAL COMMENTS	Information sharing challenges in end of life care and the potential for EPACCS EPACCS are relatively new and have the potential to greatly improve information sharing across health and social care settings, and to inform decision making out-of-hours. There is a need for more qualitative evidence to guide the roll-out and implementation of EPACCs across the UK (and in other countries), so this study is
---

timely and relevant.

However, to make sense of the findings it is important to be more clear about the context in which the data was gathered (in particular the fact that the data was collected pre-implementation of an EPACCs needs to be mentioned up front, not just in the discussion). The paper could be strengthened by discussing EPACCs in some detail (eg giving examples of where EPACCs are currently in existence, and how they have been evaluated to date).

Introduction

- Please mention examples of EPACCs eg Co-ordination My Care, the South West London EPACCs, the Leeds EPACCs, the KIS in Scotland. Petrova et al, provide an overview of different systems: https://spcare.bmj.com/content/8/4/447?ijkey=49e1dea6907fc51199fc612d610338db87321238&keytype=tf_ipsecsha.
- Qualitative evidence on epaccs has been collected in previous mixed method evaluation studies. Perhaps some of those findings could be mentioned in the Introduction. Eg. Zheng et al. 2013; Tapsfield et al 2016, Finucane et al 2019. It would be good to set this study in context of the earlier findings (though the references i mention here are specific to the Scottish KIS, there are some generalisable findings. At the same time, I agree there is a need for qualitative evidence from other settings and context, so this study is welcome and needed..)
- Perhaps also draw attention to Leniz's systematic review up front (not just in the discussion) to set this study in context of previous work.
- Page 5, Line 43. Please insert '2013' to reflect when the cost data relates to.
- Page 5, Line 50. Statement that 'patient registration on such electronic systems' is very low. I would argue that this varies across settings and regions. In Scotland, 69% of die have an electronic care co-ordination record by the time of death (Finucane et al 2019, BJGP).
- Clarify the aim, in context of other work. And shorten, instead of 'This article reports data from a project exploring...' better to just state 'The aim of our study was to.....'

Methods

- It would be interesting to include the interview schedule as supplementary/additional information.
- Some further detail on the context/setting in which the data was collected is needed. Participants were recruited from the North of England – what EPACCs are currently available or being implemented in that settings; how likely are participants to have exposure to these systems?
- Participants – how were participants recruited? From where/what setting? What was the inclusion/exclusion criteria if any?
- Some sentences may be redundant eg " We endeavoured to engage in a process of reflexivity throughout the project". This could be removed.
- There is a need for clarity regarding the number of focus groups and interviews – this has not been reported. How many focus groups were run, who participated in focus groups? How many interviews were conducted? Who participated in interviews?
- Is there any further information on patient participant characteristics? What was their primary diagnosis?
- Is the PPI rep a co-author? If not, should they be mentioned in the acknowledgements?

	Findings  • Consider using the term ‘participants/ instead of ‘interviewees. • Avoid duplication of content in the introduction and findings. The first paragraph of the findings on ‘information sharing challenges’ is similar to a paragraph in the introduction – so does not stand out as a new finding. • Some of the findings are rather obvious, eg “Inappropriate hospital admissions and paramedic call-outs....were also suggested to be a waste of scarce NHS resources....”. (p9, L28) Better to focus on new knowledge. • The main point to draw from the quote by GP6 is unclear. What is this quote illustrating? • P10, L8: “GPs are not available 24/7.....” perhaps mention primary care Out of hours in this paragraph (as a service that would also benefit from shared info). • Information sharing systems, P11, L16: perhaps add “and up to date”. • Page 12, Line 6 - better to distinguish between electronic records local to a specific general practice, hospice or care settings, and EPaCCS. • “P12, L41 “NHS information systems” – what does this refer to – which systems? • Page 14, Line 41, “Moves to facilitate information sharing through electronic records with the aim of improved care were seen as welcome and necessary progress” Please specify which participants perceived this (especially as GPs were less enthusiastic according to an earlier paragraph? • Some of the concerns raised have limited validity as the participants don’t seem to have yet experienced using EPaCCS. So perhaps rephrase, to make it clear, that these are concerns in advance of implementing an EPaCCS (and perhaps later mention who they could be addressed). Discussion  • Well written overview of findings. • For transparency, and to facilitate interpretation of the findings it is essential to also make it clear that these findings were collected prior to EPaCCS implementation (is in the the context/setting section of the methods). • Comparison with other work – there is a focus on the perceived challenges from the perspective of GPs, however, the KIS in Scotland relies heavily on GP input, but still works relatively well, and most people have a KIS record at the time of death. • The discussion suggests further qualitative research to better understand the infrastructure required to sustain an EPaCCS, as well as the benefits and challenges. Please note that there is recent evidence on this, post-implementation of epaccs, from mixed methods Scottish studies. Of course, as the authors suggest, more is needed (from different contexts and stages of implementation). • P19, L38 – please clarify what is meant by ‘equitable use of the system’ Conclusion Remove redundant words “Our study suggests that”.
--	--

VERSION 1 – AUTHOR RESPONSE

Thank you for allowing us to revise and re-submit this paper and to respond to the comments of the reviewers. We thank both reviewers for the time and thought they have put into their comments, and we are gratified that both found some merit in our work. Both have also raised some concerns, and we address those in our re-submission (as outlined below).

Reviewer	Paper section	Comment	Author Response	Evidence from manuscript
Reviewer 1				
	Abstract	You state that current systems for ACP do not work well, but it is hard to understand how an electronic version would be a solution to documentation that is fragmented and often of low quality (if completed at all). The solution surely needs to start with addressing ACP processes before putting them into a system that has yet to prove itself and that requires training and resources?	Thank you for this comment. We have included additional discussion of the need for investment into training around initiating and documenting ACP into the abstract and conclusion of the paper to highlight this issue.	Page 2
	Introduction	Page 5. Line 13 Please explain what Admiral nurses are for those outside the UK	We have added a brief description of dementia nurses.	Page 4
		Page 5. Line 50. Surely professional reluctance to initiate EOL discussions relates more to a lack of communication and lack of subsequent documentation generally, rather than a reflection of the use /uptake of EPaCCS?	We discuss some of the wider issues regarding EOL communication in the discussion and the need to address these. Clinicians' reluctance to engage in these discussions is a challenge to the introduction of an EPaCCS, without these occurring there is no information to input. As such we reflect in the paper that this means that introduction of the system must be accompanied by investment in training to improve confidence in engaging in end of life discussions.	
	Methods	I found this to be lacking the details that are described in the proforma at the end of the paper. In particular recruitment approaches / population pool /sampling and reporting on participant demographics.	We have added further detail regarding the recruitment approach for clinician and patient interviews.	Page 6
		Page 6. Line 52. Please detail numbers approached and numbers	Unfortunately we do not have records of the number of	

		finally interviewed	individuals approached and those interviewed. Family and carers were recruited through GP practices we did not request that they kept a record of those approached, we only received details of those who agreed to be contacted by the research team.	
		interviewed Page 7. Line 23. Please explain Macmillan and Marie Curie roles	We have explained what these roles are.	Page 7
		Page 7. Please detail which participants were interviewed and which were part of a focus group. Were the latter grouped via discipline and patient / family for example?	Only clinicians participated in focus groups. We have added extra detail which discussed who took part in each focus group.	Page 6
		Page 7. Line 58. Is there a reason that interview transcripts did not go back to participants for verification?	We appreciate this comment, but returning the transcripts to participants was not part of the protocol for this study.	
		Page 8. Line 12. Could you please expand on reflexivity and why it was important?	We have removed this sentence as per the suggestion of reviewer 2. However, reflexivity is discussed in the methods section of the paper.	Page 8
		Page 8. Line 18. Did you perform any cross-verification analysis? What was your level of agreement?	Three researchers coded a proportion of the transcripts, and came together and discussed any disagreements to come to a consensus as described in the data analysis section of the paper. We did not measure the level of agreement/disagreement.	Page 8
		Page 12. Line 6. Is the reader to assume that every patient at end of life in this region has an ACP?	We have amended this sentence to make it clear that not use of ACPs is variable.	Page 12
	Discussion	It would be interesting to see a summary (table?) of the findings by grouping eg, patients, paramedics, GPs. There are important considerations for each as well as competing interests. It would also be interesting to see a discussion on whether the implementation of an electronic patient case note system would in fact have any effect on ACP completion rates, quality of ACP documentation and so on. It is idealistic to think that a	Thank you for this suggestion, we have considered this but decided not to include it in the paper as we feel that the differences between groups are nuanced and we do not feel they lend themselves to presentation in a table.	

		system like this would solve all communication between disciplines as it always is dependent on users imputing the information.		
		Page 18. Line 19. This is the first mention of recruitment concerns – not mentioned previously and an important consideration in study design	We have added to the methods that patient and family member interviewees were recruited through clinician gatekeepers.	Page 6
		Page 18. Line 52. Comparison with other work. You make important points here about limitations of EPaCCS in relation to EOL management. This section addresses most of my questions and is almost lost in the paper.	Thank you for this comment. We agree that it is important these points are not lost in the paper so have reiterated in the abstract and conclusion.	Pages 2 and 19
		Page 19. Line 42 Conclusion. This is pretty limited and should reflect more the findings from the study. Page 18 lines 52 onwards is (in part) more of a conclusion that represents your work (as above).	Thank you for this comment we have amended the conclusion in light of this comment to make it stronger and highlight the importance of investment in other skills to improve end-of-life management.	Page 19
Reviewer 2				
		However, to make sense of the findings it is important to be more clear about the context in which the data was gathered (in particular the fact that the data was collected pre-implementation of an EPACCs needs to be mentioned up front, not just in the discussion). The paper could be strengthened by discussing EPACCs in some detail (eg giving examples of where EPACCs are currently in existence, and how they have been evaluated to date).	Thank you for this comment, we agree that it is important to ensure this is clear. We have stated that the data were generated pre-implementation in the article summary and in the methods section of the paper.	Pages 3 and 6
		Please mention examples of EPACCs eg Co-ordination My Care, the South West London EPACCs, the Leeds EPACCs, the KIS in Scotland.	Thank you for this suggestion. We have added this to the introduction section.	Page 5
		Qualitative evidence on epaccs has been collected in previous mixed method evaluation studies. Perhaps some of those findings could be mentioned in the Introduction. Eg. Zheng et al. 2013; Tapsfield et al 2016, Finucane et al 2019. It would be good to set this study in context of the earlier findings (though the	Thank you for this suggestion. We have increased the discussion of previous literature in the introduction section of the paper and added additional references as suggested.	Page 5

		references i mention here are specific to the Scottish KIS, there are some generalisable findings. At the same time, I agree there is a need for qualitative evidence from other settings and context, so this study is welcome and needed..)		
		Perhaps also draw attention to Leniz's systematic review up front (not just in the discussion) to set this study in context of previous work.	We have added this reference to the introduction as suggested.	Page 5
		Page 5, Line 43. Please insert '2013' to reflect when the cost data relates to.	This has been added to the document	Page 4
		Page 5, Line 50. Statement that 'patient registration on such electronic systems' is very low. I would argue that this varies across settings and regions. In Scotland, 69% of die have an electronic care co-ordination record by the time of death (Finucane et al 2019, BJGP).	Thank you for this comment. We have amended this section and added the suggested reference.	Page 5
		Clarify the aim, in context of other work. And shorten, instead of 'This article reports data from a project exploring...' better to just state 'The aim of our study was to...."	We have amended that aim as suggested.	Page 5
	Methods	It would be interesting to include the interview schedule as supplementary/additional information.	Thank you for this suggestion, an example interview schedule has been included as supplementary material.	Referred to on page 6
		Some further detail on the context/setting in which the data was collected is needed. Participants were recruited from the North of England – what EPACCS are currently available or being implemented in that settings; how likely are participants to have exposure to these systems?	Thank you for this comment, we accept that more clarity is needed throughout the paper regarding the fact that the EPaCCS was in the process of being implemented rather than already embedded in practice. We have added additional detail which states that participants were recruited from an area where a trial was being set up and reflections were hypothetical rather than direct experience of using the system.	Pages 3 and 6
		Participants – how were participants recruited? From where/what setting? What was the inclusion/exclusion criteria if any?	As per the comment of reviewer 1, we have included additional details regarding the recruitment processes.	Page 6
		Some sentences may be redundant	This sentence has been	

		eg “ We endeavoured to engage in a process of reflexivity throughout the project”. This could be removed.	removed.	
		There is a need for clarity regarding the number of focus groups and interviews – this has not been reported. How many focus groups were run, who participated in focus groups? How many interviews were conducted? Who participated in interviews?	We had added additional details regarding the number of interviews and focus groups, as well as the demographics of the clinician focus groups.	Page 6
		Is there any further information on patient participant characteristics? What was their primary diagnosis?	We have added information regarding the primary prognosis of patients participants to the paper.	Page 6
		Is the PPI rep a co-author? If not, should they be mentioned in the acknowledgements?	Michele Spencer is mentioned in the acknowledgements section of the paper, we have added addition details to make it clear that her role was as PPI rep	Page 22
	Findings	Consider using the term ‘participants/ instead of ‘interviewees.	We have made the change as suggested.	
		Avoid duplication of content in the introduction and findings. The first paragraph of the findings on ‘information sharing challenges’ is similar to a paragraph in the introduction – so does not stand out as a new finding.	We have removed this paragraph from the paper.	
		Some of the findings are rather obvious, eg “Inappropriate hospital admissions and paramedic call-outs....were also suggested to be a waste of scarce NHS resources....”. (p9, L28) Better to focus on new knowledge.	As suggested we have removed this sentence from the paper.	
		The main point to draw from the quote by GP6 is unclear. What is this quote illustrating?	We have added some extra information to this quote to make the purpose of this piece of data clearer- that GPs may be frustrated by palliative care admissions as they often possess information/knowledge that would facilitate the patient being kept at home, however this information is not available when primary care GPs are not working.	Page 9
		P10, L8: “GPs are not available 24/7.....” perhaps mention primary	An extra sentence has been added to this paragraph to	Page 9

		care Out of hours in this paragraph (as a service that would also benefit from shared info).	highlight that out of hours GPs and paramedics would benefit from shared information to facilitate their management of end of life patients.	
		Information sharing systems, P11, L16: perhaps add “and up to date”.	We have added this suggestion to document.	Page 11
		Page 12, Line 6 - better to distinguish between electronic records local to a specific general practice, hospice or care settings, and EPaCCS.	We have amended the text here to provide a clearer distinction between records.	Page 10/11
		“P12, L41 “NHS information systems” – what does this refer to – which systems?	We have clarified that we are referring to electronic patient records.	Page 12
		Page 14, Line 41, “Moves to facilitate information sharing through electronic records with the aim of improved care were seen as welcome and necessary progress” Please specify which participants perceived this (especially as GPs were less enthusiastic according to an earlier paragraph?	This finding relates to patient and family member perspectives, we have amended this section to make this clearer.	Page 14
		Some of the concerns raised have limited validity as the participants don’t seem to have yet experienced using EPaCCS. So perhaps rephrase, to make it clear, that these are concerns in advance of implementing an EPaCCS (and perhaps later mention who they could be addressed).	Thank you for this comment. We have tried to make it clearer through this section that participants were reflecting on a proposed system rather than one they were currently using.	Page 13 and 14
	Discussion	For transparency, and to facilitate interpretation of the findings it is essential to also make it clear that these findings were collected prior to EPaCCS implementation (is in the the context/setting section of the methods).	As mentioned in relation to previous comments we have added additional detail to the methods and introduction sections of the paper to make it clear that the data we are presenting was collection prior to the implementation of the EPaCCS.	
		Comparison with other work – there is a focus on the perceived challenges from the perspective of GPs, however, the KIS in Scotland relies heavily on GP input, but still works relatively well, and most people have a KIS record at the	Thank you for pointing this out. We have added a reference to the Scottish KIS to demonstrate that in practice there has been success with the approach of GPs inputting to the system.	Page 18

		time of death.		
		The discussion suggests further qualitative research to better understand the infrastructure required to sustain an EPaCCS, as well as the benefits and challenges. Please note that there is recent evidence on this, post-implementation of epaccs, from mixed methods Scottish studies. Of course, as the authors suggest, more is needed (from different contexts and stages of implementation).	As suggested by the reviewer we have included findings from the Scottish KIS to both the introduction and discussion of the paper. We have amended this section to highlight that we feel that a focus on the impacts of the system on the experience of patients and families is a particular area of need.	Pages 18 and 19
		P19, L38 – please clarify what is meant by ‘equitable use of the system’	We have edited this sentence for clarity.	Page 19
	Conclusion	Remove redundant words “Our study suggests that”.	This has been removed.	

VERSION 2 – REVIEW

REVIEWER	Deb Rawlings Flinders University, Adelaide, South Australia
REVIEW RETURNED	09-Jun-2020

GENERAL COMMENTS	Thank you for addressing the changes. I do have two minor points: 1) Not allowing the participants to review transcripts is a limitation. Please add in the methods section that this was not undertaken and consider adding this to the strengths and weaknesses section. 2) You will still need to describe reflexivity as not all readers will understand its place and importance in this process. Thank you.
--

REVIEWER	Anne Finucane Marie Curie Hospice Edinburgh, Scotland, UK
REVIEW RETURNED	16-Jun-2020

GENERAL COMMENTS	Thank-you for revising this paper in line with suggestions.
---

VERSION 2 – AUTHOR RESPONSE

Response to reviewers

We thank the Reviewers for their time in reviewing our revised manuscript. We are pleased that Reviewer 2 is now satisfied with the revisions made. Below we detail our response to the remaining minor comments from Reviewer 1 (in italics).

1) Not allowing the participants to review transcripts is a limitation. Please add in the methods section that this was not undertaken and consider adding this to the strengths and weaknesses section.

The reviewer has suggested that we should add that we did not engage in interviewee transcript review as a limitation of our study. We are aware that debate exists around the possible value of interviewee transcript review (Hagens et al. 2009 BMC Medical Research Methodology). Although we recognise that such an approach may have value for some teams and projects, as stated in the previous round of reviews this was not in the protocol for this study. Based on our own disciplinary practices and standards we do not feel that the value and credibility of our work is reduced by not following this step.

2) You will still need to describe reflexivity as not all readers will understand its place and importance in this process.

As suggested we have added extra detail describing reflexivity.

'Reflexivity is the process of accounting for the situatedness of the researcher within the research and the potential effect of this on the data collected and interpretation.'³²

VERSION 3 – REVIEW

REVIEWER	Deb Rawlings Flinders University, Adelaide, South Australia.
REVIEW RETURNED	29-Jul-2020
GENERAL COMMENTS	Thank you for your response. I do not agree with your views on participants reviewing transcripts. As a member of an ethics committee and guided by the Australian National Statement on Ethical Conduct in Human Research this speaks to respect for participants within research. However it is not mandatory so I accept that you have a very different ethical approach and do not view this as a limitation.